# A Gauge-Theory-based Graph Neural Network

## Abstract

We introduce a gauge-theoretic framework for graph neural networks on arbitrary graphs with local frames. Each directed edge carries a link variable $U_{ij} \in O(d)$ that parallel-transports node features, and the invariant head aggregates a finite, explicit dictionary of gauge invariants (open strings and Wilson loops). Using the First Fundamental Theorem for $O(d)$ on mixed tensor spaces, we prove that this dictionary generates all $O(d)$-invariant polynomials, yielding a universal approximation result on compact sets for continuous invariant targets. We further formulate learning directly on the orbit space $X/(S_n \times O(d))$ and establish a nonuniform learnability guarantee via bounded-Lipschitz slices. We realize the theory in a lightweight message-passing architecture. On a synthetic gauge diagnostic, the model attains almost perfect generalization while passing local gauge probes and maintaining numerical $S_n \times O(d)$. On the QM9 dataset, an augmented variant that includes atomic numbers and interatomic distances improves regression accuracy. These results show that a finite gauge-invariant dictionary, implemented with standard message passing, is both theoretically expressive and practically effective for symmetry-aware learning on graphs.

## 1 Introduction

Symmetry is a primary source of inductive bias in modern geometric machine learning. In many scientific settings, node-wise features should be equivariant to global rotations and reflections, and the model output should be invariant (or transformed in a prescribed way). Enforcing these properties improves data efficiency, stability, and generalization. Group-equivariant neural networks make this principle concrete for compact groups and homogeneous spaces, ranging from the Abelian case of a circle (Fourier features) to non-Abelian Lie groups underlying three-dimensional (3D) rotations and translations (Cohen & Welling, 2016; Esteves, 2020; Cohen et al., 2019; Worrall et al., 2017; Kondor & Trivedi, 2018). For graphs, permutation invariance and equivariance with respect to node relabeling is fundamental. Maron et al. characterized all linear $S_n$-invariant and $S_n$-equivariant layers via index-equality patterns and used them to build expressive GNNs. Later, they proved the universality of $G$-invariant networks (with $G \leq S_n$) when higher-order tensors are allowed (Maron et al., 2019a;b). Subsequent work extended universality to certain equivariant targets and connected the proofs to Stone–Weierstrass arguments (Keriven & Peyré, 2019).

For protein structure prediction and design, the outputs must be rigid-motion invariant, meaning that the same structure up under a global $SE(3)$ transformation should be scored identically. AlphaFold 2 achieves this via its *Invariant Point Attention* (IPA) mechanism inside the structure module, which computes geometry-aware attention in local residue frames and yields global rigid-motion invariance and equivariance (Jumper et al., 2021). Recent RosettaFold 2 variants likewise incorporate 3D equivariant modules inspired by $SE(3)$-equivariant transformers (Baek et al., 2023). These successes underscore the value of hardwiring symmetry.Beyond strict, rigid congruence, practitioners often require *task-tolerant* equivalence. Small displacements along intrinsically flexible directions (e.g., the normal modes of elastic network models) may be acceptable in some regions, whereas the catalytic sites require near-atomic fidelity. This suggests learning on the orbit space $X/G$ endowed with a *task-weighted pseudometric* that downweights deviations along dynamically soft modes and emphasizes rigid or function-critical substructures. Public resources, such as PDBj ProMode-Elastic, provide normal-mode directions and fluctuation profiles that can instantiate such weights (Protein Data Bank Japan (PDBj), 2020; Wako & Endo, 2017). Our gauge-invariant dictio-

nary (open strings and Wilson loops) in the present paper is compatible with this view: it provides a finite set of $O(d)$-invariant coordinates on $X/G$ onto which a task-specific metric can be imposed, enabling principled tolerances in training and evaluation.

In terms of expressivity, standard message-passing GNNs with local aggregation (sum/mean/max) and shared updates are *no stronger than* the 1-Weisfeiler-Leman graph isomorphism test (Xu et al., 2019; Morris et al., 2019). Consequently, they cannot distinguish many non-isomorphic graphs and fail to represent large families of permutation-invariant functions. In particular, they do *not* enjoy a universal approximation property (UAP) over continuous invariant functions on graphs without augmentations that go beyond plain message passing (e.g., higher-order tensors, subgraph lifts, or global features) (Keriven & Peyré, 2019). This gap motivates architectures that (i) enlarge the invariant feature algebra and (ii) yield a provable UAP while respecting symmetry.

In 3D spaces, models equivariant to rotations (and often translations) have become standard. Tensor Field Networks encode features in irreducible $SO(3)$ types; $SE(3)$-Transformers build self-attention layers that are equivariant to the full Euclidean group; EGNNs deliver a lightweight $E(n)$-equivariant message-passing variant; and spectral/SO(3) approaches, such as Spherical CNNs and Clebsch-Gordan Nets, operate directly in the Fourier/Wigner domain (Thomas et al., 2018; Fuchs et al., 2020; Satorras et al., 2021; Cohen et al., 2018; Kondor et al., 2018). Beyond global rotations, many problems exist in bundles with local frames, where the relevant symmetry is gauge rather than global. Gauge-equivariant CNNs generalize group-equivariance to local gauge transformations on manifolds and discrete geometries, and in lattice gauge theory (Rothe, 2005), Wilson loops are canonical gauge invariants. Recent "lattice gauge equivariant CNNs" build such Wilson-loop (Wilson, 1974) features and show exact gauge equivariance on grids; relatedly, a Wigner-Eckart parameterization yields principled kernel families for group-equivariant layers (Weiler & Cesa, 2019; Favoni et al., 2022; Lang et al., 2021).

We study graphs with node features $h_i \in \mathbb{R}^d$ and link variables $U_{ij} \in O(d)$ that transport between local frames; $r \in O(d)$ acts by $h_i \longmapsto r\,h_i,\ U_{ij} \longmapsto r\,U_{ij}\,r^{-1}$. Our goals are twofold: (i) Construct a finite, explicit dictionary of $O(d)$-invariant features that enables universal approximation of continuous invariants on compact subsets; and (ii) Place learning on the quotient/orbit space, and translate standard uniform-convergence tools to this setting. The motivation for a gauge-based formulation is that local frames arise naturally in physics and geometry; transporting features along edges via $U_{ij}$ aligns information before aggregation. Gauge invariants are generated by open strings $\langle h_i, W h_j \rangle$ (transport and inner product) and closed loops $\mathrm{tr}(W)$ (Wilson loops), where $W$ is a word in the *alphabet* $\{U_{ij}^{\pm 1}\}$. This mirrors the lattice-gauge theory, in which Wilson loops generate gauge-invariant observables and it connects directly to recent gauge-equivariant neural layers (Weiler & Cesa, 2019; Favoni et al., 2022; Lang et al., 2021).

The contributions of this study are as follows. **First**, we construct the invariant generator dictionary, composed of open strings and loops. Building on classical invariant theory, pairwise contractions for $O(d)$ and trace identities for matrices, we show that all $O(d)$-invariant polynomials in $\{h_i\}$ and $\{U_{ij}\}$ can be expressed as polynomials in two families: $s_{i_0 \dots i_m} = \langle h_{i_0}, U_{i_0 i_1} \cdots U_{i_{m-1} i_m} h_{i_m} \rangle$, $w_{i_0 \dots i_{m-1}} = \mathrm{tr}(U_{i_0 i_1} \cdots U_{i_{m-1} i_0})$, with length-0 case $s_{ij} = \langle h_i, h_j \rangle$. Our argument follows the First Fundamental Theorem for $O(d)$ on mixed tensor spaces and the identification $V^* \otimes V \simeq \mathrm{End}(V)$, together with standard matrix-invariant results (Goodman & Wallach, 2009; Procesi, 2007; Lang et al., 2021). **Second**, we show the UAP for continuous $O(d)$-invariants on compacta. Since $O(d)$ is reductive, the invariant ring is finitely generated, and the polynomials in our dictionary separate orbits and include constants. According to the Stone–Weierstrass theorem on the compact orbit space, multilayer perceptrons (MLPs) applied on top of the dictionary achieve uniform universal approximation of continuous $O(d)$-invariant targets. For permutation symmetry, the same statement holds after symmetrization, consistent with the $S_n$ universality line (Maron et al., 2019b;a).) **Third**, we formalize quotient-space learning and generalization. Training can be conducted either on $X$ with a $G$-invariant law or directly on $X/G$ via the pushforward measure; the risks coincide with those of invariant hypotheses. This viewpoint cleanly justifies data augmentation by group actions and clarifies why symmetry constraints often reduce effective complexity, echoing observations from equivariant GNNs and group-CNNs (Cohen & Welling, 2016; Cohen et al., 2019; Esteves, 2020). **Finally**, we argue the practical realization as message passing. We demonstrate the realization of open-string and loop features within a message-passing pipeline, providing an architecture that complements the prior permutation-equivariant bases and

$G_c$-equivariant layers (Maron et al., 2019a; Thomas et al., 2018; Fuchs et al., 2020; Satorras et al., 2021; Cohen et al., 2018; Kondor et al., 2018). Our construction unifies strands from permutation-equivariant GNN theory (linear bases and universality), 3D $SE(3)/SO(3)$-equivariant models, and gauge-equivariant CNNs (Maron et al., 2019b;a; Thomas et al., 2018; Fuchs et al., 2020; Satorras et al., 2021; Cohen et al., 2018; Kondor et al., 2018; Weiler & Cesa, 2019; Favoni et al., 2022; Lang et al., 2021). Compared with Maron et al.'s $S_n$ universality, we augment the symmetry with $O(d)$ and introduce link variables that transform by conjugation, yielding an explicit, finite invariant dictionary on general graphs. Our formulation targets arbitrary graphs and recovers Wilson-loop-style invariants as a special case (Maron et al., 2019b;a; Weiler & Cesa, 2019; Favoni et al., 2022).

## 2 RELATED WORK

**Permutation-equivariant graph networks.** Maron et al. characterized all *linear* $S_n$-invariant/equivariant layers by tying weights according to index-equality patterns, yielding a complete linear basis for tensors on graphs (Maron et al., 2019a). They further proved the universality of $G$-invariant networks (with $G \leq S_n$) when higher-order tensorization was permitted (Maron et al., 2019b). These studies clarified that permutation symmetry can be imposed without loss of expressivity, but universality may require high tensor orders, with attendant memory and computational costs. Moreover, the theory is purely discrete ($S_n$) and does not address continuous internal symmetries such as $O(d)$ acting on vector features, nor *gauge* variables on edges. We augment the symmetry from $S_n$ to $S_n \times O(d)$ and introduces edge-wise group elements $U_{ij}$ transforming by conjugation, while retaining a finite, constructive dictionary of invariant features.

**Euclidean/$SO(3)$-equivariant models in 3D.** Tensor Field Networks encode features in irreducible $SO(3)$ types and compose them via tensor products (Thomas et al., 2018), and the SE(3)-Transformer brings equivariant self-attention to point clouds and molecules (Fuchs et al., 2020). EGNN proposes a lightweight $E(n)$-equivariant message passing architecture using distances and coordinate updates (Satorras et al., 2021). Spectral designs such as Spherical CNNs and Clebsch-Gordan Nets operate directly in the noncommutative Fourier/Wigner domain on $SO(3)$ (Cohen et al., 2018; Kondor et al., 2018). These models convincingly demonstrate the sample-efficiency benefits of equivariance in 3D tasks; however, they typically (i) rely on coordinates or global frames rather than edgewise transports, (ii) do not furnish a finite, explicit generating set for *all $O(d)$*-invariant polynomials on graphs, and (iii) provide limited theory on learnability on the orbit space. We address (i)-(iii) by working with the *gauge* link variables and proving the UAP result.

**Gauge-equivariant convolutions and Wilson loops.** Gauge-equivariant CNNs generalize group equivariance to local frames on manifolds and discrete geometries (Weiler & Cesa, 2019). In lattice gauge theory, Wilson loops-traces of parallel transport around closed loops—form canonical gauge-invariant observables; recent "lattice gauge-equivariant CNNs" build such features and show exact gauge equivariance on grids (Favoni et al., 2022). However, these constructions are tailored to regular lattices or specific meshes and do not provide a general finite generating set of invariants for arbitrary graphs. Our formulation extends the Wilson-loop/open-string picture to arbitrary graphs with link variables $U_{ij} \in O(d)$, and we prove that the resulting dictionary generates fully invariant polynomial algebra, which, in turn, yields a universal approximation theorem on compact subsets.

**Quotient feature spaces and Reynolds networks.** Sannai et al. introduce the *quotient feature space* (QFS) to analyze generalization of invariant/equivariant DNNs via the geometry of the orbit space, deriving bounds whose leading term scales with the "volume" of the QFS (Sannai et al., 2021). For permutation symmetry, their analysis yielded substantial improvements (for example, DeepSets gains a $\sqrt{n!}$ factor) and provides optimal approximation rates for invariant targets. Complementarily, Sannai et al. proposed *Reynolds Networks* (ReyNets), which implement invariant/equivariantization using a *reductive Reynolds operator* that averages over a carefully chosen *Reynolds design*, reducing the cost of averaging over the entire group (Weyl, 1946) (e.g., from $O(n!)$ for $S_n$) to near-quadratic for graphs (Sannai et al., 2024). In addition, Kumagai & Sannai established a unified universal approximation theorem for equivariant maps using Group CNNs (Kumagai & Sannai, 2020). Methodologically, our approach differs in that we avoid group integration by constructing a *finite $O(d)$-invariant generator dictionary* and proving UAP on compacta. Nevertheless, the QFS viewpoint and Reynolds-based constructions are compatible with our *orbit-space learning*.

**General theory of group-equivariant CNNs.** A unified representation theoretic view has emerged, with kernels parameterized via irreducibles and Clebsch–Gordan structures (Cohen & Welling, 2016; Cohen et al., 2019; Kondor & Trivedi, 2018; Lang et al., 2021; Esteves, 2020; Worrall et al., 2017). Conceptually, this rests on Peter–Weyl and harmonic analyses. Our theoretical ingredients are complementary: we use the First Fundamental Theorem (FFT) (Goodman & Wallach, 2009) for $O(d)$ on mixed tensors together with matrix trace identities (Procesi, 1976; 2007) to obtain a *finite* invariant generator set on graphs (open strings $\langle h_i, W h_j \rangle$ and Wilson loops $\mathrm{tr}(W)$), and then apply Stone–Weierstrass theorem on the orbit space to achieve UAP.

**Remaining issues and our positioning.** Across the above families one encounters (a) universality that requires high tensor orders or bandwidths, (b) the lack of a finite, explicit generator set for joint permutation *and* continuous group invariants on graphs with edge transports, and (c) limited treatment of learnability on the quotient/orbit space. Our study addresses these issues by: (i) Constructing a finite, explicit dictionary that generates all $O(d)$-invariant polynomials in the node features and link variables, and extends to $S_n \times O(d)$ by symmetrization. (ii) Proving the UAP for continuous invariant targets on compact subsets without resorting to high-order tensorization. (iii) Formulating the empirical risk minimization (ERM) directly on the orbit space.

**Rotation/rigid-motion handling in protein models.** AlphaFold 2 introduced *Invariant Point Attention* (IPA), a geometry-aware attention mechanism that operates in local frames to produce outputs invariant (or equivariant) to global rigid motions (Jumper et al., 2021; Yang et al., 2023). RosettaFold 2 adopts related $SE(3)$-equivariant components to achieve competitive accuracy with improved scaling (Baek et al., 2023). These designs demonstrate that enforcing $SE(3)$ symmetry stabilizes the learning of 3D biomolecular data. However, they do not provide a finite, explicit *generating set of invariants* for arbitrary graphs with edge transports, nor do they formalize learning directly on orbit space with task-weighted tolerances. Our work addresses these gaps: we provide a finite generator dictionary for $S_n \times O(d)$-invariants on general graphs, and propose an orbit-space viewpoint in which rigidity or flexibility can be encoded via data-driven weights (e.g. normal-mode profiles from ProMode-Elastic (Protein Data Bank Japan (PDBj), 2020; Wako & Endo, 2017)).

## 3 Notations and formulation

We fix a graph with $n$ vertices (indices $i, j, k \in \{1, \ldots, n\}$) and a finite directed edge set $E \subset \{1, \ldots, n\}^2$. Let $G_c$ be a compact Lie group acting linearly on a real $d$-dimensional representation space $V \simeq \mathbb{R}^d$ (e.g. $G_c = O(d)$ or $SO(d)$), and let $S_n$ act by permuting the vertex labels. We write $G := S_n \times G_c$ for the fully symmetric group. In layer $\ell$, each vertex carries a *feature* $h_i^{(\ell)} \in V$, and each directed edge $j \rightarrow i$ carries a *link variable* $U_{ij}^{(\ell)} \in G_c$ (discrete parallel transport). We also allow fixed gauge-invariant edge scalars $s_{ij} \in \mathbb{R}^p$ (distances, weights, etc.), which remain unchanged by group action. The symbol $\oplus$ denotes a permutation-invariant aggregator (e.g., sum/mean/max).

For $(\sigma, r) \in S_n \times G_c$ and a state $(U, h) = \big((U_{ij})_{(i,j) \in E}, (h_i)_{i=1}^n\big)$, the action is

$$(\sigma, r) \cdot (U, h) = \Big( \big(r\, U_{\sigma^{-1}(i)\, \sigma^{-1}(j)}\, r^{-1}\big)_{(i,j) \in E}, \; \big(r\, h_{\sigma^{-1}(i)}\big)_{i=1}^n \Big). \tag{1}$$

Each layer $F^{(\ell)} : (U^{(\ell)}, h^{(\ell)}) \mapsto (U^{(\ell+1)}, h^{(\ell+1)})$ is defined by shared maps

$$\psi^{(\ell)} : V \times V \times \mathbb{R}^p \to V, \qquad \phi^{(\ell)} : V \times V \to V, \qquad \sigma^{(\ell)} : V \times V \times G_c \times \mathbb{R}^p \to G_c,$$

and a permutation-invariant aggregator $\oplus$, via

$$m_{ij}^{(\ell)} = \psi^{(\ell)}\big(U_{ij}^{(\ell)} h_j^{(\ell)}, \; h_i^{(\ell)}, \; s_{ij}\big), \quad M_i^{(\ell)} = \bigoplus_{j \neq i} m_{ij}^{(\ell)},$$

$$h_i^{(\ell+1)} = \phi^{(\ell)}\big(h_i^{(\ell)}, \; M_i^{(\ell)}\big), \quad U_{ij}^{(\ell+1)} = \sigma^{(\ell)}\big(h_i^{(\ell)}, \; h_j^{(\ell)}, \; U_{ij}^{(\ell)}, \; s_{ij}\big).$$

We assume $\psi^{(\ell)}, \phi^{(\ell)}, \sigma^{(\ell)}$ are $G_c$-equivariant (with respect to the left action on $V$ and the conjugation on $G_c$), and are shared across edges and vertices. Because the maps are shared and $\oplus$ is order-independent, each layer is $G$-equivariant: $F^{(\ell)}\big((\sigma, r) \cdot (U, h)\big) = (\sigma, r) \cdot F^{(\ell)}(U, h) \; \forall (\sigma, r) \in$

$S_n \times G_c$. Consequently, any "depth-$L$ trunk" $T_L := F^{(L-1)} \circ \cdots \circ F^{(0)}$ is $G$-equivariant. Let us write $(U^{(L)}, h^{(L)}) = T_L(U, h)$. For any word $W$ in the alphabet $\{U_{ij}^{\pm 1} : (i, j) \in E\}$, define $s_{i,j;W}(U, h) := \langle h_i, W h_j \rangle$ with $\langle \cdot, \cdot \rangle$ being the inner product, and $w_W(U) := \mathrm{tr}(W)$, including the length-0 case $s_{i,j;\mathrm{id}} = \langle h_i, h_j \rangle$. We form a dictionary of $O(d)$-invariant generators:

$$\Phi^{(L)}(U^{(L)}, h^{(L)}) = \left( s_{i,j;W}^{(L)} = \langle h_i^{(L)}, W h_j^{(L)} \rangle, \quad w_W^{(L)} = \mathrm{tr}(W) \right)_{(i,j),\, W},$$

where $W$ ranges over words in the alphabet $\{ U_{ab}^{(L)\,\pm 1} \}$. Using a permutation-invariant aggregation SymAgg over indices (e.g., sum/mean), the final prediction is $\widehat{y} = \rho\Big( \mathrm{SymAgg}\big(\Phi^{(L)}(U, h)\big) \Big)$, with $\rho$ an MLP which takes its value in an output space $\mathcal{Y}$ typically embedded into some $\mathbb{R}^q$. Since $T_L$ is $G$-equivariant and $\Phi^{(L)}$ is $O(d)$-invariant (and becomes $S_n$-invariant after SymAgg), the composition yields an $S_n \times O(d)$-invariant $\widehat{y}$. For $x \in \mathbb{R}^d$, $\|x\|_2 = \big( \sum_k x_k^2 \big)^{1/2}$. For $A \in \mathbb{R}^{m \times n}$, $\|A\|_F = \big( \sum_{i,j} a_{ij}^2 \big)^{1/2} = \sqrt{\mathrm{tr}(A^\top A)}$, and the spectral norm is $\|A\|_{2\to 2} = \sup_{\|x\|_2 = 1} \|Ax\|_2$.

# 4 MAIN RESULTS

## 4.1 UNIVERSAL APPROXIMATION PROPERTY

Although we can formulate our GNN by using any compact Lie groups, we provide the UAP with its specific case here; $G_c = O(d)$. We fix $d, n \in \mathbb{N}$ and a finite directed edge set $E \subset \{1, \ldots, n\}^2$. Let $X = (\mathbb{R}^d)^n \times O(d)^E$ with elements $x = (U, h)$, where $h = (h_1, \ldots, h_n)$, $h_i \in \mathbb{R}^d$, and $U = (U_{ij})_{(i,j) \in E}$, $U_{ij} \in O(d)$. Let $\mathcal{P}(X)$ denote a set of polynomials on $X$; write $\mathcal{P}(X)^G$ for the $G$-invariant subring. The group $G_c = O(d)$ acts as in (1).

**Theorem 4.1.** *Let $\mathcal{A}$ be the $\mathbb{R}$-algebra generated by all $s_{i,j;W}$ and $w_W$. Then, $\mathcal{A} = \mathcal{P}(X)^{O(d)}$. In particular, $\mathcal{P}(X)^{O(d)}$ is finitely generated; hence, a finite tuple $\Phi(x) = (\phi_1(x), \ldots, \phi_M(x))$ with $\phi_m \in \mathcal{A}$ $(m = 1, 2, \ldots, M)$ exists, such that every $O(d)$-invariant polynomial is a polynomial in $\phi_1, \ldots, \phi_M$. Moreover, the generators can be chosen using words whose lengths are bounded by a constant depending only on $d$ and $|E|$.*

**Theorem 4.2.** *Suppose further that there exist parameters for which $F^{(\ell)}$ is the identity: $\psi^{(\ell)} \equiv 0$, $\phi^{(\ell)}(h, M) = h$, $\sigma^{(\ell)}(h_i, h_j, U_{ij}, s_{ij}) = U_{ij}$. Let $K \subset X$ be compact and $O(d)$-invariant. Then, for every $f \in C(K)^{O(d)}$ and every $\varepsilon > 0$, there exist an integer $L \geq 0$, trunk $T_L$, and a continuous readout $N : \mathbb{R}^M \to \mathbb{R}$ (e.g., an MLP with a nonpolynomial activation) such that*

$$\sup_{x \in K} \big| f(x) - N\big(\Phi(T_L(x))\big) \big| < \varepsilon.$$

*In particular, because the identity trunk $T_0 = \mathrm{id}$ is contained in the class by assumption,*

$$\overline{\{ N \circ \Phi \mid N \in C(\mathbb{R}^M) \}}^{\, \|\cdot\|_{C(K)}} = C(K)^{O(d)}.$$

*Therefore, our network can approximate any continuous $O(d)$-invariant function of the* initial $(U, h)$. *The same statement holds for $G = S_n \times O(d)$ after symmetrizing the indices (or aggregating).*

## 4.2 LEARNING ON THE ORBIT SPACE

Let $G := S_n \times G_c$ act on $X$ by (1), and let $\pi : X \to X/G$ be the quotient map (Lee, 2012). For any data distribution $\mu$ on $X$, write $\bar{\mu} := \pi_\# \mu$ for its pushforward to $X/G$. Assume labels are orbit-constant, i.e., there exists $\bar{y} : X/G \to \mathcal{Y}$ with $y(x) = \bar{y}(\pi(x))$. The next lemma enables us to define the risk on the orbit space.

**Lemma 4.3.** *For any $G$-invariant predictor $f$ with $\bar{f}$ such that $\bar{f} \circ \pi = f$ and any Lipschitz loss $\tilde{l} : \mathcal{Y} \times \mathcal{Y} \to \mathbb{R}$, we have $\mathbb{E}_{(x,y) \sim \mu}\big[ \ell(f(x), y) \big] = \mathbb{E}_{(\bar{x}, \bar{y}) \sim \bar{\mu}}\big[ \ell(\bar{f}(\bar{x}), \bar{y}(\bar{x})) \big]$.*

Define the induced invariant hypotheses on the orbit space

$$\mathscr{H}_L := \Big\{ \bar{f} : X/G \to \mathcal{Y} \,\Big|\, \bar{f} = \rho \circ \mathrm{SymAgg} \circ \Phi^{(L)} \circ T_L \circ s, \ s \text{ any Borel section of } \pi \Big\}.$$

**Theorem 4.4.** *Let $\tilde{l} : \mathcal{Y} \times \mathcal{Y} \to [0, 1]$ be $L_{\tilde{l}}$-Lipschitz in its arguments, and define $\bar{l}(\bar{f}; \bar{x}, \bar{y}) \equiv \tilde{l}(\bar{f}(\bar{x}), \bar{y})$. Then, $\mathscr{H}_L$ is nonuniformly learnable on $\bar{\mu}$ with respect to $\bar{l}$.*

*Remark 4.5.* Appendix C defines nonuniform learnability and proves Theorem 4.4.

# 5 EXPERIMENTS

We evaluate our gauge-equivariant message-passing architecture on two settings: (i) a proprietary dataset, which we call *synthetic gauge diagnostic* hereafter (details omitted for anonymity), and (ii) molecular property regression on **QM9** (Ramakrishnan et al., 2014; Ruddigkeit et al., 2012) with compact training splits. Preprocessing and splits follow MoleculeNet (Wu et al., 2018). Unless noted, all models use the same trunk and invariant readout described below.

## 5.1 SETUP

**Model.** We employed the same model for two datasets mentioned above. Nodes carry vectors $h_i \in \mathbb{R}^3$ and directed edges carry link variables $U_{ij} \in O(3)$ that parallel-transport features. Each layer transports neighbor features into the local frame and forms a small invariant message:

$$a_{ij} = U_{ij}h_j, \quad b_i = h_i, \quad \mathrm{inv}_{ij} = \left( \|a_{ij}\|^2, \|b_i\|^2, \langle a_{ij}, b_i \rangle, 1 \right),$$

$$m_{ij} = \alpha(\mathrm{inv}_{ij})\, a_{ij} + \beta(\mathrm{inv}_{ij})\, b_i, \quad M_i = \sum_{j \neq i} m_{ij}, \quad h_i^{\mathrm{new}} = \gamma(\mathrm{inv}_i)\, h_i + \delta(\mathrm{inv}_i)\, M_i,$$

$$d_i := \left| \{ j : (j,i) \in E \} \right|, \qquad \mathrm{inv}_i := \left( \|h_i\|^2, \frac{1}{d_i}\sum_{j \neq i} \|a_{ij}\|^2, \frac{1}{d_i}\sum_{j \neq i} \langle a_{ij}, h_i \rangle, 1 \right).$$

The averages are 0 when $d_i = 0$; $(\alpha, \beta)$ and $(\gamma, \delta)$ are small scalar MLPs on invariants. In the fast runs we do *not* update link. The invariant readout aggregates $S_0 = \sum_{i \leq j} \langle h_i, h_j \rangle$, $S_1 = \sum_{i,j} \langle h_i, U_{ij}h_j \rangle$, $W_3 = \sum_{i \neq j \neq k} \mathrm{tr}(U_{ij}U_{jk}U_{ki})$, and the holonomy energy $E_3 = \sum_{i \neq j \neq k} \|I - U_{ij}U_{jk}U_{ki}\|_F^2$, then feeds $[S_0, S_1, W_3, E_3]$ to an MLP head. In all experiments we treat $E_3$ as an invariant feature fed into the readout MLP together with $[S_0, S_1, W_3]$. Note that $E_3$ does *not* enlarge the invariant dictionary used in our UAP: for each triangle $\Omega_{ijk} = U_{ij}U_{jk}U_{ki}$, $\|I - \Omega_{ijk}\|_F^2 = 2d - 2\,\mathrm{tr}(\Omega_{ijk})$. We include $E_3$ as an engineered feature that can ease optimization.

**Training protocol.** We employ the Adam optimizer (Kingma & Ba, 2015) (learning rate $3 \times 10^{-4}$), with a weight decay of $10^{-5}$, SiLU (Swish) activation (Elfwing et al., 2018; Ramachandran et al., 2017), batch size 512; depth $L$ is set per configuration (see tables/figures). For the *synthetic gauge diagnostic* (binary) we train with binary cross-entropy (BCE) on logits and report accuracy. For QM9, we standardize targets; a normalized MSE=1.0 corresponds to predicting the dataset mean.

**Graph construction.** For QM9, we build a $k$NN molecular graph in 3D: each atom connects to its $k$ nearest neighbors ($k \in \{6, 8\}$ in our runs). For the synthetic gauge diagnostic, the graph topology is fixed (6-node ring with an extra triangle), so $k$ does not apply.

**Variants.** *Gauge-GNN (base)* uses only $(U, h)$ and the invariant dictionary $[S_0, S_1, W_3, E_3]$. For QM9, *Gauge-GNN+* augments the scalar inputs with chemistry cues: per-atom atomic numbers $Z_i$ and pairwise distances $d_{ij} = \|x_i - x_j\|_2$ (from QM9 geometries). For each edge we concatenate $(Z_i, Z_j, d_{ij})$ to $\mathrm{inv}_{ij}$, and the readout can include simple symmetric sums over $Z$ and distances. The trunk and invariant head are unchanged; depth $L$ and $k$-NN degree $k$ are chosen per configuration.

**Checkpoint selection.** We train for $T$ epochs and evaluate the validation MSE after each epoch. We then select the checkpoint $\hat{t} = \arg\min_{t \in \{1, \dots, T\}} \mathrm{MSE}_{\mathrm{val}}(t)$ and report all test metrics *once* at epoch $\hat{t}$. The test split is never used for training or tuning. When comparing configurations (depth, $k$, and the "Gauge-GNN+" variant), the same protocol is applied independently to each configuration.

**Sanity checks.** After each run, a random global rotation $R \in O(3)$ is applied; features are rotated and links conjugated $(U, h) \mapsto (RUR^{-1}, Rh)$, and the resulting prediction differences are recorded.

## 5.2 SYNTHETIC GAUGE DIAGNOSTIC

**Task.** We perform *binary classification* on graphs endowed with node vectors and gauge links. The two classes are: (NEGATIVE) globally *flat* connection, and (POSITIVE) connection with a small *localized holonomy* injected on a few triangles.

| Label | $\theta$ (rad) | $k_\triangle$ | $\sigma$ | Depth | Params | Task |
|--------|------|------|------|------|------|------|
| Strong | 0.4 | 1 | **0.0** | 2 | 9673 | binary |
| Medium | 0.4 | 1 | **0.2** | 2 | 9673 | binary |
| Weak | 0.4 | 1 | **0.4** | 2 | 9673 | binary |

Table 1: **Difficulty tiers in the synthetic diagnostic and models.** Higher $\sigma$ is harder.

| Run (dir) | Epoch | Val BCE $\downarrow$ | Test BCE $\downarrow$ | Test Acc $\uparrow$ | Inv. sanity $\downarrow$ ($|z(r) - z(\cdot)|$) |
|--------|------|------|------|------|------|
| Strong | 40 | 0.000 | 0.006 | 0.999 | $1.91 \times 10^{-6}$ |
| Medium | 53 | 0.001 | 0.007 | 0.999 | $9.54 \times 10^{-7}$ |
| Weak | 80 | 0.038 | 0.038 | 1.000 | $1.83 \times 10^{-5}$ |

Table 2: **Original synthetic (binary).** Trained with BCE; we report accuracy and a global $O(3)$ invariance drift $\Delta_{\mathrm{glob}} := \mathrm{median}_{r \in SO(3)} |z(r) - z(\cdot)|$ on the validation set, where $z(\cdot)$ is the scalar logit. Lower is better. Local (node-wise) gauge drift is reported separately.

**State space.** We fix $n = 6$ nodes and dimension $d=3$. Each node carries a vector $h_i \in \mathbb{R}^d$ and each directed edge $j \to i$ carries an orthogonal link $U_{ij} \in O(d)$ (one per direction).

**How we generate a sample.** Given $(\theta, k_\triangle, \sigma)$, we vary:

(i) **Flat background.** Concretely, we control difficulty via $(\theta, k_\triangle, \sigma)$ where $\theta$ is the injected holonomy angle, $k_\triangle$ the number of perturbed triangles, and $\sigma$ the node-feature noise scale. Unless otherwise stated we fix $\theta = 0.4$ rad and $k_\triangle = 1$ and vary only $\sigma$: **Strong** ($\sigma = 0.0$), **Medium** ($\sigma = 0.2$), **Weak** ($\sigma = 0.4$). Larger $\theta$ or $k_\triangle$ increases signal (easier), while larger $\sigma$ reduces SNR (harder). We give the details of the original data in Table 1. Draw random local frames $Q_i \in SO(d)$ and set a flat connection $U_{ij} \leftarrow Q_i Q_j^\top$ (so $U_{ji} = U_{ij}^\top$).

(ii) **Node features.** Sample $h_i \sim \mathcal{N}(0, \sigma^2 I_d)$ independently.

(iii) **Positive examples (holonomy injection).** Choose $k_\triangle$ distinct triangles $(i, j, k)$. On each, pick one edge and left-multiply by a small rotation $R(\theta) = \exp(\theta K)$ with a random skew generator $K$ of unit Frobenius norm: $U_{ab} \leftarrow R(\theta) U_{ab}$, and keep $U_{ba} = U_{ab}^\top$.

We label NEGATIVE when no injection is applied, and POSITIVE otherwise. All datasets are **balanced** (50% positives, 50% negatives).

**Difficulty tiers.** We vary only the feature noise $\sigma$ while fixing $\theta=0.4$ rad and $k_\triangle=1$: **Strong** ($\sigma=0.0$), **Medium** ($\sigma=0.2$), **Weak** ($\sigma=0.4$). Higher $\sigma \Rightarrow$ lower SNR $\Rightarrow$ harder.

**Model/training.** The trunk transports neighbor features $a_{ij} = U_{ij} h_j$ and forms invariant scalars $\mathrm{inv}_{ij} = (\|a_{ij}\|^2, \|h_i\|^2, \langle a_{ij}, h_i \rangle, 1)$ to gate vector messages; links are kept fixed in the fast runs. The invariant head aggregates $S_0 = \sum_{i \leq j} \langle h_i, h_j \rangle$, $S_1 = \sum_{i,j} \langle h_i, U_{ij} h_j \rangle$, $W_3 = \sum_{i \neq j \neq k} \mathrm{tr}(U_{ij} U_{jk} U_{ki})$, and $E_3 = \sum_{i \neq j \neq k} \|I - U_{ij} U_{jk} U_{ki}\|_F^2$, then feeds $[S_0, S_1, W_3, E_3]$ to a small MLP. We train with **BCE on logits** and report accuracy (train/val/test where logged).

**Sanity probes.** (i) *Global $O(3)$ invariance:* sample $r \in O(3)$ and measure $\big|\mathrm{logit}(U, h) - \mathrm{logit}(rUr^{-1}, rh)\big|$ (reported for the synthetic binary task in Table 2); (ii) *Local gauge invariance:* sample independent $\{Q_i\} \subset O(3)$ and measure $\Delta_g = |\hat{y}(U, h) - \hat{y}(QUQ^{-1}, Qh)|$ (summary statistics in Table 4; for binary we compute drift on the *logit*, for QM9 on the normalized scalar output); (iii) *Orthogonality drift:* monitor $\|U_{ij}^\top U_{ij} - I\|_F$ during training. We do not tabulate (iii), but across finished runs violations remained $\lesssim 10^{-7}$.

We follow the above protocol (validation-based checkpoint selection; invariant readout). Figures 1 and 2 show accuracy and loss curves. We observe stable optimization and near-perfect generalization under all three difficulty levels. Table 2 summarizes the best validation-epoch metrics.

**Takeaways.** (*i*) The compact invariant dictionary enables fast, stable learning across settings. (*ii*) The invariance probe confirms numerical $O(3)$ equivariance without group averaging. (*iii*) Task difficulty can be dialed via the generation parameters (used here as weak/medium/strong).

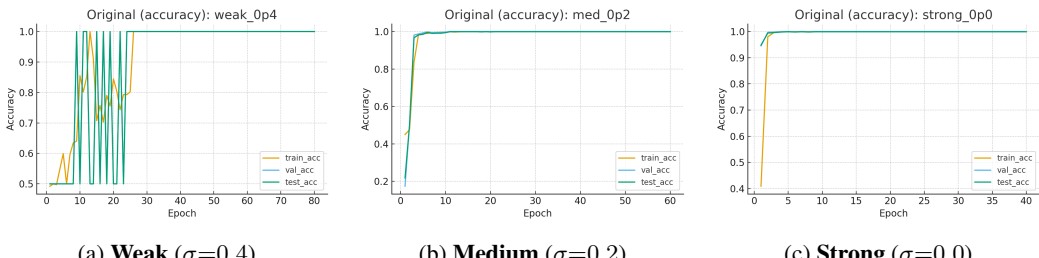

(a) **Weak** ($\sigma$=0.4)  (b) **Medium** ($\sigma$=0.2)  (c) **Strong** ($\sigma$=0.0)

Figure 1: **Original-data synthetic diagnostic: accuracy vs epoch.** Three noise levels with $\theta$=0.4 rad, $k_\triangle$=1 fixed.

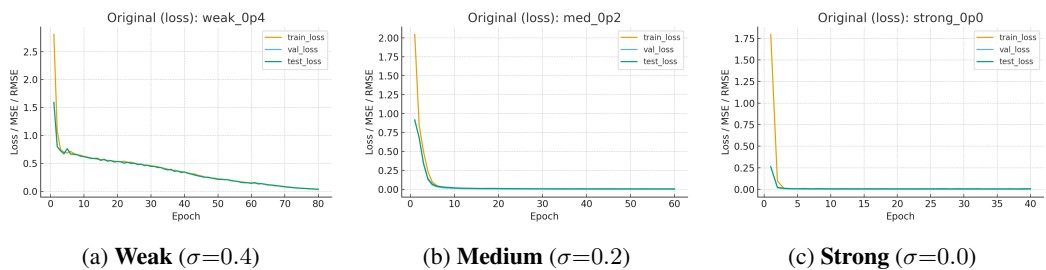

(a) **Weak** ($\sigma$=0.4)  (b) **Medium** ($\sigma$=0.2)  (c) **Strong** ($\sigma$=0.0)

Figure 2: **Synthetic gauge diagnostic: BCE loss vs epoch.** Train/val/test where logged; difficulty controlled by feature noise $\sigma$.

## 5.3 RESULTS ON QM9

We use a compact QM9 split of $3000/300/300$ molecules for train/val/test. We evaluate two standard QM9 properties: (i) the molecular dipole moment magnitude $\mu$ (units: Debye), and (ii) the electronic internal energy at $0\,\mathrm{K}$, $U_0$ (units: Hartree $E_\mathrm{h}$), both computed at B3LYP/6-31G(2df,p) for the equilibrium geometry (Ramakrishnan et al., 2014; Wu et al., 2018). Targets are standardized using *train*-split statistics, $\tilde{y} = (y - \mu_\mathrm{train})/\sigma_\mathrm{train}$. Unless noted, all regression metrics are reported on this normalized scale. For the dipole $\mu$ we additionally report a physical RMSE by rescaling: $\mathrm{RMSE_{phys}} = \sigma_\mathrm{train} \cdot \mathrm{RMSE_{norm}}$ (in Debye; $1\,\mathrm{D} = 3.33564 \times 10^{-30}\,\mathrm{C \cdot m} \approx 0.20819\,e\!\cdot\!\text{Å}$).

**Quantitative results.** Table 3 reports best-by-validation results per target. Gauge-GNN+ consistently improves regression quality. For dipole moment $\mu$, test MSE drops from $0.984$ to $0.484$ (normalized RMSE $0.696$, physical RMSE $\approx 1.046$). For $U_0$, test MSE drops from $1.257$ to $0.225$

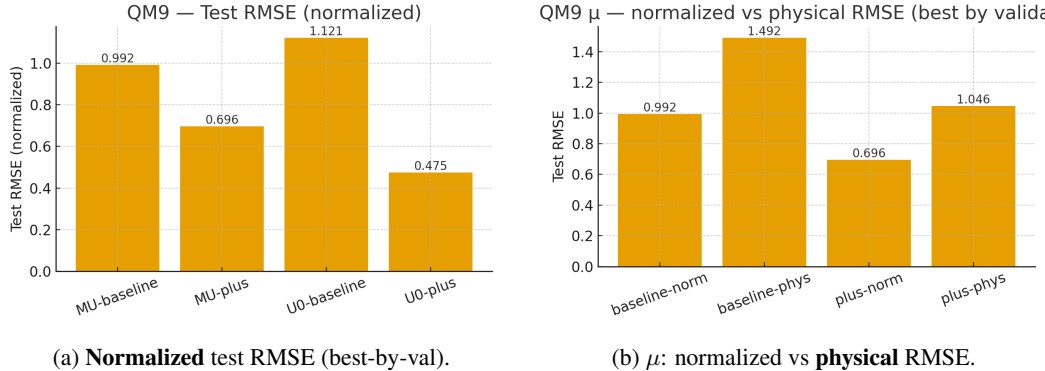

(a) **Normalized** test RMSE (best-by-val).  (b) $\mu$: normalized vs **physical** RMSE.

Figure 3: **QM9 regression. Left:** normalized test RMSE comparing Gauge–GNN vs Gauge–GNN+ (Z, distances) for $\mu$ and $U_0$ (selection by validation MSE). **Right:** dipole $\mu$—normalized vs physical RMSE (physical = normalized $\times \sigma_y$, $\sigma_y \approx 1.5035$ Debye).

| Target | Model | Depth | $k$ | Epoch | Val MSE | Test MSE | Test RMSE |
|--------|-------|-------|-----|-------|---------|----------|-----------|
| $\mu$ (dipole) | Gauge-GNN (base) | 1 | 6 | best | 0.787 | 0.984 | 0.992 / 1.492 |
| $\mu$ (dipole) | Gauge-GNN+ (Z, dist) | 2 | 8 | 30 | 0.543 | 0.484 | 0.696 / 1.046 |
| $U_0$ | Gauge-GNN (base) | 1 | 6 | best | 0.961 | 1.257 | 1.121 |
| $U_0$ | Gauge-GNN+ (Z, dist) | 2 | 8 | 23 | 0.222 | 0.225 | 0.475 |

Table 3: **QM9 regression (small split, best by validation).** Test RMSE is reported on the normalized scale; for $\mu$ we also show physical RMSE after rescaling by $\sigma_y \approx 1.5035$. Gauge-GNN+ augments inputs with atomic number $Z$ and interatomic distances.

| Setting | Mean drift | Std | Max |
|---------|-----------|-----|-----|
| Binary (weak, depth 2) | $4.42\times10^{-5}$ | $2.06\times10^{-3}$ | $1.74\times10^{-1}$ |
| QM9 base ($\mu$, depth 0, $k{=}6$) | $2.97\times10^{-3}$ | $2.30\times10^{-3}$ | $6.62\times10^{-3}$ |
| QM9 base ($U_0$, depth 0, $k{=}6$) | $1.16\times10^{-2}$ | $8.99\times10^{-3}$ | $3.31\times10^{-2}$ |
| QM9 + ($\mu$, depth 1, $k{=}6$) | $1.53\times10^{-5}$ | $1.39\times10^{-5}$ | $4.73\times10^{-5}$ |
| QM9 + ($U_0$, depth 2, $k{=}8$) | $3.73\times10^{-9}$ | $5.27\times10^{-9}$ | $1.49\times10^{-8}$ |

Table 4: **Local gauge probe.** Absolute output drift $\Delta_{\mathrm{g}} = |\hat{y}(U,h) - \hat{y}(U',h')|$ under node-wise $Q_i \in O(3)$ with $h'_i = Q_i h_i$ and $U'_{ij} = Q_i U_{ij} Q_j^{-1}$. For the binary task we measure drift on the logit $z$; for QM9, we measure drift in the normalized target units. We sample 20 random gauges per example on the synthetic diagnostic and 10 on QM9; we report mean, std., and max across examples and trials.

(normalized RMSE 0.475). Figure 3a visualizes normalized RMSE for $\mu$ and $U_0$, and Figure 3b compares normalized vs. physical RMSE for $\mu$.

### 5.4 LOCAL GAUGE PROBE

To verify invariance beyond global $O(d)$, we apply independent node-wise frame changes $Q_i \in O(3)$ and transform inputs by $h'_i = Q_i h_i$, $U'_{ij} = Q_i U_{ij} Q_j^{-1}$, and recompute the prediction and record the absolute drift $\Delta_{\mathrm{g}} = |\hat{y}(U,h) - \hat{y}(U',h')|$. Across runs (Table 4), the *Gauge-GNN+* exhibits drifts at the $10^{-5}$–$10^{-9}$ level, effectively invariant; the base QM9 model shows small but nonzero drift ($\sim 10^{-3}$–$10^{-2}$), and the synthetic binary task is invariant on average with rare outliers (max $\approx 1.7\times10^{-1}$ in logit units), which occur near the decision boundary and are mitigated by using double precision or reorthogonalizing $U_{ij}$. These measurements corroborate that our invariant head (open strings + loops) cancels local frame choices, i.e., learning effectively occurs on the orbit space.

## 6 CONCLUDING REMARKS

We proposed a gauge-theoretic GNN with a *finite* invariant dictionary (open strings, Wilson loops) that, by the FFT for $O(d)$ on mixed tensors plus trace identities, generates all $O(d)$-invariant polynomials, yielding UAP on compact sets. We also cast ERM on the orbit space and proved nonuniform learnability under bounded-Lipschitz assumptions. A lightweight instantiation performs strongly on a synthetic gauge diagnostic and improves QM9 regression while passing global/local gauge probes. Limitations include compact domains and small, $O(d)$-only experiments. Next steps: (i) extend to a wider range of Lie groups (e.g., $SO(3)$) and conduct larger-scale experiments; (ii) learnable, re-orthogonalized links (with holonomy regularization); (iii) scalable invariant estimation via subgraph/word-length sampling; and (iv) sharper orbit-space generalization rates. Overall, a finite dictionary plus standard message passing is both expressive and practical for symmetry-aware graph learning.

**Reproducibility.** Code, configs, and scripts are provided in the anonymous artifact `iclr2026-artifact-fixed.zip` (supplementary). All runs use Python 3.10/PyTorch 2.1 on a single A100 (40GB), seeds {1,2,3}. Splits: synthetic as in Sec. 5.2; QM9 3000/300/300 (MoleculeNet preprocessing).

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

# A   USE OF LARGE LANGUAGE MODELS (LLMs)

We used ChatGPT for language polishing of author-written text and for drafting small helper scripts (e.g., data plotting, gauge-probe harness). All code and results were reviewed, run, and validated by the authors. All scientific ideas, model designs, proofs, and claims are the authors' work; the LLM is not an author.

# B   PROOF OF THEOREMS 4.1 AND 4.2

## B.1   PROOF OF THEOREM 4.1

We consider, the ring $T_{k,h,t}$ of $GL(n)$-invariants of $k$ $n \times n$ matrices, $h$ $n$-vectors, and $t$ $n$-covectors.

**Definition B.1** (Procesi's bracket (polarized determinant))**.** Let $V \simeq \mathbb{R}^n$ be a finite-dimensional real vector space with a fixed basis.

**(i) Bracket of vectors.**   For $v_1, \dots, v_n \in V$, define

$$[v_1, \dots, v_n] \ := \ \det\big(\begin{bmatrix} v_1 & \cdots & v_n \end{bmatrix}\big),$$

i.e., the determinant of the $n \times n$ matrix whose columns are the coordinates of the $v_i$. Equivalently, for a fixed nonzero volume form $\omega \in \wedge^n V^*$,

$$[v_1, \dots, v_n] \ = \ \omega(v_1 \wedge \cdots \wedge v_n).$$

**(ii) Bracket of covectors.**   For $\phi_1, \dots, \phi_n \in V^*$, define

$$[\phi_1, \dots, \phi_n] \ := \ \det\begin{pmatrix} \phi_1^\top \\ \vdots \\ \phi_n^\top \end{pmatrix},$$

the determinant of the $n \times n$ matrix whose rows are the coordinate rows of the $\phi_i$. Equivalently, for $\Omega \in \wedge^n V$ with $\langle \omega, \Omega \rangle = 1$,

$$[\phi_1, \dots, \phi_n] \ = \ (\phi_1 \wedge \cdots \wedge \phi_n)(\Omega).$$

*Remark* B.2 (Transformation law and invariance). For $g \in GL(V)$,

$$[gv_1, \dots, gv_n] \ = \ \det(g)\,[v_1, \dots, v_n], \qquad [g^{-\top}\phi_1, \dots, g^{-\top}\phi_n] \ = \ \det(g)^{-1}\,[\phi_1, \dots, \phi_n].$$

Hence the vector bracket is $SL(V)$-invariant (a $GL(V)$ semi-invariant of weight $+1$), and the covector bracket has weight $-1$. In particular, for $g \in O(n)$ we have $\det(g) = \pm 1$, so the bracket is invariant under $SO(n)$ but flips sign under reflections; it is therefore *not $O(n)$-invariant* unless squared or otherwise combined to kill the sign.

**Definition B.3.** Let $X_1, \dots, X_k \in \mathrm{End}(V)$ be matrix variables, $v_i \in V$ vectors, and $\phi_j \in V^*$ covectors. For a word (monomial) $M$ in the $X_\ell$, define

$$\langle \phi_j, \ Mv_i \rangle, \qquad \big[M_1 v_{i_1}, \ \dots, \ M_n v_{i_n}\big], \qquad \big[M_1^\top \phi_{i_1}, \ \dots, \ M_n^\top \phi_{i_n}\big].$$

Procesi's theorem (Procesi, 1976, Thm. 12.1) states that, together with the matrix-only invariants (traces of words $\mathrm{tr}\,W(X)$), these generate the $GL(V)$-invariant ring on the space of $k$ matrices, $h$ vectors, and $t$ covectors:

**Theorem B.4.** *$T_{k,h,t}$ is generated by the following elements:*

(i) *Invariants of $k$ matrices alone;*

(ii) *Scalar products $\langle \phi_j, Mv_i \rangle$, where $M$ is a monomial in the given matrices, $\phi_j$ a covector, $v_i$ a vector;*

(iii) *Brackets $[M_1 v_{i_1}, M_2 v_{i_2}, \dots M_n v_{i_n}]$, where $M_i$'s are monomials in the matrices and the $v_j$'s are vectors.*

(iv) *Brackets* $[M_1^\top \phi_{i_1}, M_2^\top \phi_{i_2}, \ldots M_n^\top \phi_{i_n}]$, *where $M_i$'s are monomials in the matrices and the $\phi_j$'s are covectors.*

*Remark* B.5 (Link to this paper ($O(d)$-invariants: open strings and loops)). In our setting the symmetry group is $O(d)$, and the metric identifies $V \simeq V^*$. On the orthogonal locus $U^\top U = I$ we have $U^\top = U^{-1}$. Then Procesi's items (a)(b) specialize to

$$\underbrace{\mathrm{tr}\, W(U, U^{-1})}_{\text{closed loops / Wilson loops}} \quad , \qquad \underbrace{\langle h_i,\, W(U, U^{-1})\, h_j \rangle}_{\text{open strings / transported pairings}} \quad ,$$

which generate the $O(d)$-invariant polynomial algebra we use. By contrast, the bracket generators in (iii)(iv) are $SO(d)$-invariant but change sign under reflections, so they are not $O(d)$-invariant and can be omitted from our minimal generating set. For background on the FFT and contractions for $O(V)$, see Goodman & Wallach (2009, Cor. 5.2.3 and §5.3).

We also have the first fundamental theorem for $O(d)$ Procesi (1976, Thm. 7.1) :

**Theorem B.6.** *Every orthogonal invariant of $i$ matrices $(A_1, \ldots, A_i)$ is a polynomial in the elements $\mathrm{tr}(U_{i_1}, U_{i_2}, \ldots, U_{i_k})$, where $U_j = A_j$, or $U_j = A_j^\top$.*

Now, set $X = (\mathbb{R}^d)^n \times O(d)^E$ and $G_c = O(d)$ which acts on $X$ by

$$r \cdot (U, h) = \big((rU_{ij}r^{-1})_{(i,j) \in E}, rh\big).$$

Let us define the invariant open strings and loops:

$$s_{i,j;W}(U, h) = \langle h_i, Wh_j \rangle, \quad w_W(U) = \mathrm{tr}(W) \quad (r \in O(d)),$$

for words $W$ in the "alphabet" $U_{ij}^{\pm 1}$, and let $\mathcal{A}$ be the $\mathbb{R}$-algebra they generate; $\mathcal{A} \subset \mathcal{P}(X)^{O(d)}$. In fact, since $U_{ij}^{-1} = U_{ij}^\top$ for $U_{ij} \in O(d)$, $s_{i,j;W}$'s and $w_W(U)$'s are polynomials in the entries of $(h, U)$, and

$$s_{i,j;W}(rUr^{-1}, rh) = \langle rh_i, rWr^{-1}\, rh_j \rangle = \langle h_i, Wh_j \rangle, \quad w_W(rUr^{-1}) = \mathrm{tr}(rWr^{-1}) = \mathrm{tr}(W),$$

so they are $O(d)$-invariant. Since $O(d)$ preserves $\langle \cdot, \cdot \rangle$, we identify $V \simeq V^*$ and $V^* \otimes V \simeq \mathrm{End}(V)$. By the first fundamental theorem for $O(d)$ (Goodman & Wallach, 2009; Procesi, 1976; 2007), every $O(d)$-invariant polynomial in $(U, h)$ is a polynomial in open strings $\langle h_i, Wh_j \rangle$ and Wilson loop traces $\mathrm{tr}(W)$ with $W$ a word in $\{U_{ij}, U_{ij}^\top\}$. Hence, the invariant ring $\mathcal{P}(X)^{O(d)}$ is generated by $\{s_{i,j;W}, w_W\}$, i.e. $\mathcal{A} = \mathcal{P}(X)^{O(d)}$. Since $O(d)$ is reductive, Hilbert's finiteness theorem implies that $\mathcal{P}(X)^{O(d)}$ is finitely generated. This means that there exist invariant polynomials $\Phi = (\phi_1, \phi_2, \ldots, \phi_M)$, where each $\phi_i$ is a polynomial in $s_{i,j;W}$ and $w_W$, such that every $O(d)$-invariant polynomial is a polynomial in $\phi_1, \phi_2, \ldots, \phi_M$. Moreover, by Cayley-Hamilton and the Razmyslov-Procesi trace identities (Procesi, 2007), any word $W$ in $\{U_{ij}, U_{ij}^\top\}$ reduces to a linear combination of words of length $\leq C(d)$ (in particular, $U_{ij}^k$ is a polynomial in $I, U_{ij}, \ldots, U_{ij}^{d-1}$). Thus the generators may be chosen with uniformly bounded word length depending only on $d$.

$\square$

## B.2 Proof of Theorem 4.2

Before proving Theorem 4.2, we first prepare the Stone–Weierstrass theorem. Here, $C(X, R)$ denotes a set of functions on a space $X$.

**Theorem B.7.** *Suppose $X$ is a compact Hausdorff space and $A$ is a subalgebra of $C(X, R)$, which contains a non-zero constant function. Then, $A$ is dense in $C(X, R)$ if and only if it separates points.*

Now, let $\pi : K \to K/G$ be the quotient map ,and define $\Phi : K \to \mathbb{R}^M$. For compact linear actions, invariant polynomials separate orbits. That is, $\Phi$ is constant on each $G$-orbit and induces a continuous embedding $\iota : K/G \to \Phi(K) \subset \mathbb{R}^M$ by $\iota([x]) = \Phi(x)$ (see, Figure 4).

By noting that $K/G$ is compact and $\Phi(K)$ is Hausdorff, $\iota(\cdot)$ is a homeomorphism from $K/G$ to $\Phi(K)$; thus, we can take its inverse $\iota^{-1}$.

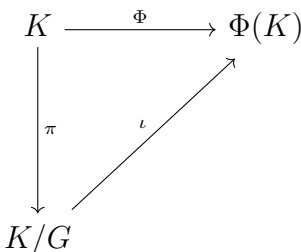

Figure 4: A path diagram.

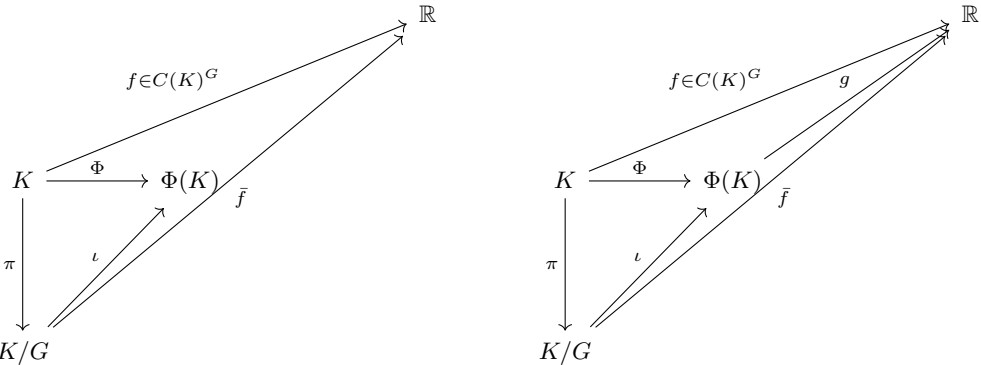

(a) Orbit-space factorization via $\Phi = \iota \circ \pi$ and $f = \bar{f} \circ \pi$.

(b) Introducing $g$ so that $f = g \circ \Phi$.

Figure 5: **Two equivalent path diagrams.** Left: factorization through the orbit space. Right: factorization through the invariant coordinate map $\Phi$.

Let us take a $G$-invariant continuous function $f \in C(K)^G$, which induces $\bar{f} : K/G \to \mathbb{R}$ (see, Figure 5(a)). Then, take $g \equiv \bar{f} \circ \iota^{-1} : \Phi(K) \to \mathbb{R}$. There exists a unique continuous function on $\Phi(K)$, $g \in C(\Phi(K))$, such that $f = g \circ \Phi$ (see, Figure 5(b)).

Because $g$ satisfies the assumptions of Theorem B.7, there exists a polynomial $p$ on $\Phi(K)$ satisfying

$$\sup_{z \in \Phi(K)} |g(z) - p(z)| < \varepsilon.$$

By representing $z = \Phi(x)$ with $x \in K$, we have that

$$\sup_{x \in K} |g(\Phi(x)) - p(\Phi(x))| < \varepsilon.$$

$\square$

*Remark* B.8. To enforce $S_n$-invariance we symmetrize indices (sum/mean). Since $S_n$ is finite, this coincides with the Reynolds projection onto $S_n$-invariants.

*Remark* B.9 (Relation to equivariant UAP by Group CNNs.). Kumagai & Sannai (2020) provided a unified UAP for equivariant maps realized by Group CNNs across broad group settings, including non-compact cases. Our result is complementary: we target graphs endowed with edge transports $U_{ij} \in O(d)$ and produce an explicit *finite* generator dictionary (open strings and loops) that separates orbits; UAP then follows from Stone–Weierstrass on the compact orbit space. Thus, our proof technique and model class differ, but both works support the broader thesis that symmetry constraints are compatible with expressivity.

## C  LEARNABILITY:PROOF OF Theorem 4.4

Here, we define the terms of learnability. Let $\mathcal{X}$ and $\mathcal{Y}$ be input and output spaces, respectively. Given a hypothesis set $\mathcal{H}$ and a loss function $l(\cdot; \cdot, \cdot) : \mathcal{H} \times \mathcal{X} \times \mathcal{Y} \to \mathbb{R}$ which is defined by

$l(h; x, y) = \tilde{l}(h(x), y)$ with a function $\tilde{l} : \mathcal{Y} \times \mathcal{Y} \to \mathbb{R}$, we introduce a set $l \circ \mathcal{H} \equiv \{\tilde{l}(h(\cdot), \cdot) | h \in \mathcal{H}\}$. Let us define risk $L_D(h)$ with a loss function $l(\cdot)$, hypothesis $h \in \mathcal{H}$, and a general hypothesis set $\mathcal{H}$ by

$$L_D(h) = E_{z \sim \mathcal{D}}\Big[l(h; z)\Big], \tag{2}$$

where $\mathcal{D}$ is an unknown data-generating distribution, defined on $\mathcal{Z} \equiv \mathcal{X} \times \mathcal{Y}$. The notation $z \sim \mathcal{D}$ indicates that a random variable $z = (x, y) \in \mathcal{Z}$ is drawn from $\mathcal{D}$ (i.e., $x \in \mathcal{X}$ and $y \in \mathcal{Y}$). We also use the notation $S \sim \mathcal{D}^m$ to denote that a dataset $S$ of sample size $m$ is i.i.d drawn from $\mathcal{D}$. The loss function $l(\cdot)$ depends on both $z \in \mathcal{Z}$ and $h \in \mathcal{H}$. A Bayes optimal hypothesis is any minimizer of (2) over $\mathcal{H}$. However, we do not typically know the actual distribution $\mathcal{D}$. For this reason, we minimize a surrogate quantity, namely the "empirical risk":

$$L_S(h) \equiv \frac{1}{m} \sum_{i=1}^{m} l(h; x_i, y_i),$$

where $S = \{(x_i, y_i)\}_{i=1}^{m} \sim \mathcal{D}^m$. This framework is called *empirical risk minimization* (ERM) (Vapnik, 2000). The law of large numbers indicates that $L_S(h)$ converges in probability to $L_D(h)$ as $m \to +\infty$ for each $h$. To evaluate the "goodness" of the training data, we define the following concept (Shalev-Shwartz & Ben-David, 2014). In the following, we use the notation $\mathcal{H}$ to denote the general hypothesis set.

**Definition C.1.** Let $\varepsilon$ be a positive, real number. A training set $S$ is called $\varepsilon$-*representative* with respect to the domain $\mathcal{Z} \equiv \mathcal{X} \times \mathcal{Y}$, the hypothesis set $\mathcal{H}$, the loss function $l(\cdot)$, and the distribution $\mathcal{D}$ if the following holds.

$$\big|L_S(h) - L_\mathcal{D}(h)\big| \leq \varepsilon \quad \forall h \in \mathcal{H}.$$

To determine the conditions under which the ERM scheme works well, we require the following definitions (refer to (Shalev-Shwartz & Ben-David, 2014), Definition 4.3).

**Definition C.2.** A hypothesis set $\mathcal{H}$ is said to possess the *uniform convergence property* with respect to domain $\mathcal{Z}$ and loss function $l(\cdot)$ if there exists a function $m_{\mathcal{H}}^{UC} : (0, 1)^2 \to \mathbb{N}$, called the *sample complexity*, such that for every $\varepsilon, \delta \in (0, 1)$ and every probability distribution $\mathcal{D}$ over $\mathcal{Z}$, if $S$ is a sample of $m \geq m_{\mathcal{H}}^{UC}(\varepsilon, \delta)$ elements that are drawn i.i.d. from $\mathcal{D}$, then with a probability of at least $1 - \delta$, $S$ is $\varepsilon$-representative.

**Definition C.3.** A hypothesis set $\mathcal{H}$ is said to be *nonuniformly learnable* if there exists a learning algorithm $A$ that maps dataset $S$ to a hypothesis $A(S) \in \mathcal{H}$, and a function $m_{\mathcal{H}}^{NUL} : (0, 1)^2 \times \mathcal{H} \to \mathbb{N}$, such that for every $\varepsilon, \delta \in (0, 1)$ and every $h \in \mathcal{H}$, if $m \geq m_{\mathcal{H}}(\varepsilon, \delta, h)$, then for every distribution $\mathcal{D}$ over $\mathcal{Z}$, with a probability of at least $1 - \delta$ over the choice of $S \sim \mathcal{D}^m$, we ensure that

$$L_\mathcal{D}(A(S)) \leq L_\mathcal{D}(h) + \varepsilon.$$

**Theorem C.4.** *Let $\mathcal{H}$ be a hypothesis set that can be written as a countable union of individual hypothesis sets.*

$$\mathcal{H} = \bigcup_{j \in \mathbb{N}} \mathcal{H}_j,$$

*where each $\mathcal{H}_j$ exhibits a uniform convergence property. Then, $\mathcal{H}$ is nonuniformly learnable.*

It is known that if a hypothesis set $\mathcal{H}$ satisfies nonuniform learnability, we can find a hypothesis that attains a small difference between the true and empirical risks using the structural risk minimization (SRM) principle (Shalev-Shwartz & Ben-David, 2014; Vapnik, 2000). We cite the following lemma (Dudley, 1999). Here, $I^d = [0, 1]^d$, where $d \in \mathbb{N}$.

**Lemma C.5.** *Let $d \in \mathbb{N}$, $\bar{K} > 0$, and $\overline{\mathcal{H}}_{1, \bar{K}}(I^d)$ be as follows:*

$$\overline{\mathcal{H}}_{1, \bar{K}}(I^d) = \left\{ f \in C(I^d) \Big| \sup_{x \in I^d} |f(x)| + \sup_{x \neq x'} \frac{|f(x) - f(x')|}{\|x - x'\|_{\mathbb{R}^d}} \leq \bar{K} \right\}.$$

*Then, $\overline{\mathcal{H}}_{1, \bar{K}}(I^d)$ is a Glivenko–Cantelli class for any probability measure $\mathcal{D}$ on $I^d$.*

*Remark* C.6. Regarding the definition of Glivenko–Cantelli class , see Dudley (1999).

**Nonuniform learnability via bounded-Lipschitz slices (real outputs)**    Note that the output space $\mathcal{Y}$ is embedded in $\mathbb{R}^q$. For $\bar{K}, B > 0$ define the bounded-Lipschitz slice

$$\mathcal{H}_{\bar{K},B} := \left\{ f = (f_1, \ldots, f_q) = \rho \circ \mathrm{SymmAgg} \circ \Phi_L \Big| \max_{1 \leq i \leq q} \|f_i\|_\infty \leq B, \ \max_{1 \leq i \leq q} \mathrm{Lip}(f_i) \leq \bar{K} \right\},$$

where $\|f_i\|_\infty$ and $\mathrm{Lip}(f_i)$ denote the supremum norm and the Lipschitz coefficient of $f_i$, respectively.

We have the following lemma.

**Lemma C.7** (Trunk Lipschitz). *Let $K \subset (\mathbb{R}^d)^n \times O(d)^E$ be compact. Suppose each layer $F^{(\ell)}$ is $L_\ell$-Lipschitz on $K$ (with respect to a fixed product norm), i.e., $\|F^{(\ell)}(x) - F^{(\ell)}(x')\| \leq L_\ell \|x - x'\|$ for all $x, x' \in K$. Then, the depth-L trunk $T_L := F^{(L-1)} \circ \cdots \circ F^{(0)}$ is $\left( \prod_{\ell=0}^{L-1} L_\ell \right)$-Lipschitz on $K$. If links are not updated in the fast runs (i.e., $\sigma^{(\ell)}(h_i, h_j, U_{ij}, s_{ij}) = U_{ij}$), the corresponding factors from $\sigma^{(\ell)}$ can be omitted from the product.*

*Proof.* Lipschitz constants multiply under composition: $\|T_L(x) - T_L(x')\| \leq \prod_{\ell=0}^{L-1} L_\ell \|x - x'\|$. If a component is the identity (no link update), its factor is 1. $\qquad\square$

**Lemma C.8** (Invariant head Lipschitz). *Fix compact $K \subset (\mathbb{R}^d)^n \times O(d)^E$. The map*

$$(U^{(L)}, h^{(L)}) \longmapsto (S_0, S_1, W_3, E_3)$$

*with*

$$S_0 = \sum_{i \leq j} \langle h_i, h_j \rangle, \quad S_1 = \sum_{i,j} \langle h_i, U_{ij} h_j \rangle, \quad W_3 = \sum_{i \neq j \neq k} \mathrm{tr}(U_{ij} U_{jk} U_{ki}),$$

$$E_3 = \sum_{i \neq j \neq k} \|I - U_{ij} U_{jk} U_{ki}\|_F^2$$

*is Lipschitz continuous on $K$. Consequently, $\mathrm{SymAgg} \circ \Phi^{(L)}$ is Lipschitz continuous; for the* mean *aggregator the Lipschitz constant is 1, and for the* sum *aggregator it is $\leq C(n, |E|)$, the number of summed terms (a graph-dependent constant).*

*Proof.* On $K$, vectors and orthogonal matrices are uniformly bounded ($\|U_{ij}\|_2 = 1$, $\|U_{ij}\|_F = \sqrt{d}$). Each term is a polynomial (or multilinear) expression in $(U, h)$ composed with inner product, trace, and matrix multiplication, all of which are smooth and hence Lipschitz on compact sets. Finite sums preserve Lipschitzness, with the constant growing at most linearly in the number of terms ($C(n, |E|)$). The mean aggregator is 1-Lipschitz; the sum has operator norm equal to the count of terms. $\qquad\square$

Regarding the learnability on the orbit space, we also have:

**Lemma C.9** (Lipschitzness of our predictors). *Let us equip $X/G$ with any compatible quotient metric, or with the pseudo-metric induced by the invariant dictionary. Then, each $\bar{f} = (\bar{f}_1, \ldots, \bar{f}_q) \in \mathcal{H}_L$ is bounded and Lipschitz continuous.*

*Proof.* It can be shown by the estimation of the form:

$$\mathrm{Lip}(\bar{f}_i) \leq \underbrace{\mathrm{Lip}(\rho_i)}_{\text{readout}} \cdot \underbrace{\mathrm{Lip}(\mathrm{SymAgg})}_{=1 \text{ (mean) or } \leq C(n) \text{ (sum)}} \cdot \underbrace{\mathrm{Lip}(\Phi^{(L)})}_{\text{polynomial}} \cdot \prod_{\ell=0}^{L-1} \underbrace{\mathrm{Lip}\left(F^{(\ell)} \circ \pi^{-1}\right)}_{\text{trunk}}.$$

$\qquad\square$

Based on Lemmas C.5–C.9, we have the nonuniform learnability of our model.

**Theorem C.10.** *Let $\tilde{\ell} : \mathcal{Y} \times \mathcal{Y} \to [0, 1]$ be $L_{\tilde{\ell}}$-Lipschitz in its arguments. For budgets $B_j, K_j \uparrow \infty$, define the bounded-Lipschitz slices*

$$\mathcal{H}_{B_j, K_j} := \left\{ \bar{f} \in \mathcal{H}_L : \max_{1 \leq i \leq q} \|\bar{f}_i\|_\infty \leq B_j, \ \max_{1 \leq i \leq q} \mathrm{Lip}(\bar{f}_i) \leq K_j \right\}.$$

*Then, each loss slice $\bar{\ell} \circ \mathcal{H}_{B_j, K_j}$ forms a Glivenko–Cantelli class.*

Now, we are in a position to prove Theorem 4.4. Note that $\mathscr{H}_L = \bigcup_{j \geq 1} \mathscr{H}_{K_j, B_j}$ with nondecreasing budgets $K_j, B_j \uparrow \infty$. Apply Lemma C.9 to obtain that each $\bar{l} \circ \mathscr{H}_{K_j, B_j}$ forms a Glivenko–Cantelli class. So, $\mathscr{H}_{K_j, B_j}$ possesses the uniform convergence property with respect to the loss function $\bar{l}$. But $\mathscr{H}_L = \bigcup_{K \in \mathbb{N}} \mathscr{H}_{K_j, B_j}$, and thus, $\mathscr{H}_L$ is nonuniformly learnable with respect to the loss function $\bar{l}$. $\qquad\square$

## D   OPTIONAL CURVATURE REGULARIZATION

For an oriented triangle $i \to j \to k \to i$, the discrete holonomy is $\Omega_{ijk} = U_{ij} U_{jk} U_{ki} \in O(d)$. A Yang–Mills–style penalty that promotes local flatness is

$$\mathcal{L}_{\text{flat}} \;=\; \lambda \sum_{(i,j,k) \in \mathcal{C}_3} \big\| I - \Omega_{ijk} \big\|_F^2,$$

with a chosen set of oriented 3-cycles $\mathcal{C}_3$ and weight $\lambda \geq 0$. This penalty is gauge-invariant and vanishes iff the connection is flat on the sampled cycles. Using $\|I - \Omega\|_F^2 = 2d - 2\operatorname{tr}(\Omega)$, one sees $\mathcal{L}_{\text{flat}}$ is an affine function of Wilson-loop traces and thus compatible with our invariant dictionary.

**Usage in this paper.**   We *did not* use $\mathcal{L}_{\text{flat}}$ in the main runs ($\lambda = 0$). The holonomy energy $E_3 = \sum_{i \neq j \neq k} \|I - U_{ij} U_{jk} U_{ki}\|_F^2$ appearing in the readout is an *invariant feature*, not a training penalty.

**Implementation note.**   On sparse triangle lists, $\mathcal{L}_{\text{flat}}$ is cheap to compute and differentiable. It can be used as a regularizer when learning $U_{ij}$ or when one wishes to bias toward (nearly) flat connections.

## E   REMARK: REYNOLDS PROJECTION VS. FINITE INVARIANT DICTIONARY.

A classical route to enforce invariance/equivariance is the *Reynolds operator* (group averaging) (Goodman & Wallach, 2009): for a representation space $\mathcal{F}$ and a compact group $G$, the projection $P_{\text{inv}} f := \int_G g \cdot f \, d\nu(g)$ maps any $f \in \mathcal{F}$ to the invariant subspace, where $\nu(\cdot)$ is an invariant measure. However, naïvely this entails averaging over the whole group (e.g. $O(n!)$ for $S_n$), which is prohibitive on large graphs. Reynolds Networks (ReyNets) (Sannai et al., 2024) mitigate the cost by averaging over a carefully designed finite set (a *Reynolds design*), achieving exact invariance/equivariance with near-quadratic complexity in $n$ for permutation symmetry. In contrast, our approach *circumvents group integration* for the continuous symmetry $O(d)$ by constructing an explicit, *finite* generator dictionary of invariant coordinates on $X$: open strings $\langle h_i, W h_j \rangle$ and Wilson loops $\operatorname{tr}(W)$ (with words $W$ in $\{U_{ij}^{\pm 1}\}$). By the First Fundamental Theorem and trace identities, these finitely many generators (with word lengths bounded via Cayley–Hamilton) generate the full algebra $\mathcal{P}(X)^{O(d)}$, enabling invariant heads. The two viewpoints are complementary: for the discrete factor ($S_n$), design-based averaging via ReyNet provides an efficient exact symmetrization; for the continuous factor ($O(d)$), polynomial generators obviate expensive integration while still yielding UAP on compacta via Stone–Weierstrass.

*Optional.* We considered a Yang-Mills-style flatness penalty on triangle holonomies (see App. D), but set $\lambda = 0$ in all reported experiments. Note: the holonomy energy $E_3$ used in the invariant head is a *feature*, not a loss term.

