# OpenReview forum: "A Gauge-Theory-based Graph Neural Network"
_ICLR.cc/2026/Conference — Submitted to ICLR 2026_

### Official Review · Reviewer_rJ9b · 2025-10-23

**Soundness:** 2
**Presentation:** 1
**Contribution:** 2
**Rating:** 4
**Confidence:** 3

**Summary:**

The paper proposes a theoretical framework for O(d)- and permutation-invariant GNNs, proving universal approximation for continuous invariant functions using finitely generated polynomial features. It formalizes learning on the orbit space, providing statistical guarantees for invariant hypothesis classes.

**Strengths:**

- The paper has a rigorous mathematical foundation and shows the UAP.
- Connects the orbit-space formulation to invariant learning

**Weaknesses:**

- The paper is mathematically dense and difficult to follow for readers unfamiliar with gauge theory or Wilson loops. Can the authors provide visualizations or diagrams showing how the concepts translate to graphs and network structures?
- In Section 3, are the node features limited to attributed graphs, or are 3D vertex coordinates also explicitly modeled? If coordinates are used, how are they incorporated into the network?
- Theorem 4.1 refers to generators expressed using ‘words’—can the authors clarify what these words are in terms of the network operations or polynomial functions?
- How are the directed edges in the graph constructed? Are they derived directly from the molecular structure, or defined differently for the network?
- It would strengthen the paper to include comparisons with other methods, for instance, on QM9 property prediction, as done in https://arxiv.org/abs/2406.03164 and https://arxiv.org/abs/2405.15429. Does this formulation lead to measurable performance gains over existing invariant/equivariant GNNs?

**Questions:**

See Weaknesses

---

### Official Review · Reviewer_Ph3c · 2025-10-23

**Soundness:** 2
**Presentation:** 1
**Contribution:** 2
**Rating:** 2
**Confidence:** 4

**Summary:**

This paper describes the task of fitting geometric graphs with gauge information. In these graphs, nodes possess vector information $\{h_i\} \subset \mathbb{R}^d$, and edges are associated with gauge information $\{U_{ij}\} \subset O(d)$ for parallel transport between nodes. For the group $O(d)$, the authors construct the fundamental invariants for geometric graphs with a gauge structure. Using these invariants, they derive a network architecture capable of uniformly approximating any $O(d)$-equivariant function on a compact set.

**Strengths:**

1. The author introduces concepts from gauge theory in quantum field theory, applying the ideas of "strings" and "loops" to construct invariants on geometric graphs with a gauge structure. This demonstrates the author's strong theoretical foundation in invariant theory.

2. Paper [1] similarly discussed the construction of $O(d)$ invariants for geometric graphs and also used FFT for $O(d)$. However, [1] did not address the link variables $U_{ij}$. The author's work supplements this by providing a discussion of geometric graphs with a gauge structure that incorporates these link variables.

Reference:

[1] Villar, Soledad, et al. "Scalars are universal: Equivariant machine learning, structured like classical physics."

**Weaknesses:**

1. Significant room for improvement in writing and organization. The main text contains substantial redundancy. Given the page limits for a conference paper, the introduction and related work sections need to be condensed, retaining only the content essential for reader comprehension while removing abstract details. For example, there are long descriptions of protein structure prediction and design in the introduction and related work, but these topics are barely mentioned in the theoretical and experimental sections. This content is superfluous. Furthermore, the description of contributions is overly verbose, and the third contribution is very abstract and likely to confuse readers. I suggest merging or removing it.

2. Lack of necessary motivation for the problem setup. When introducing geometric graphs with a gauge structure, the author fails to provide suitable examples. For readers with a physics background, the significance of discretizing a continuous gauge field is unclear. For readers without this background, the problem is more severe, as they will be completely unaware of why this concept is being introduced and will not know the physical meaning of the vectors $h_i$ and link variables $U_{ij}$. Even by the experimental section, the reader has not developed an understanding of this content and has no intuitive grasp of the problem's significance. I hope the author can carefully supplement this information, as it is crucial for reader comprehension.

3. Incomplete experimental setup and lack of essential comparisons. For experiment (ii), the QM9 dataset is familiar to researchers in equivariant neural networks. However, in the experiments on this classic dataset, the author only compares the results of their two proposed models, without comparing against classic models like EGNN [1], TFN [2], etc. In the experimental setup, the author does not explain what the vectors $h_i$ and link variables $U_{ij}$ correspond to, nor do they justify the necessity of using a geometric graph with a gauge structure to model molecular data.

4. Abrupt introduction of theory. The theoretical part of Section 4.2 is too brief. The definition of the terms "nonuniformly learnable," used in Theorem 4.4, is placed in the appendix. To help the reader understand this theorem in the main text, the author should dedicate some space in the main body to provide a necessary description.

5. Insufficient grasp of the paper's focus. In the introduction, the author emphasizes the connection to graph neural networks, which should be a point of great interest to GNN theory researchers. However, the subsequent discussion of $S_n$ invariance is inadequate. In Theorem 4.2, the author simply states that the discussion of an approximation theorem for $S_n \times O(d)$ only requires "symmetrizing the indices." However, practical methods for achieving this symmetrization are non-trivial. The transition from $O(d)$ invariance to $S_n \times O(d)$ in model design requires careful discussion. As the author notes in the appendix, a naive Reynolds averaging over $S_n$ has a time complexity of $O(n!)$, and techniques like Reynolds Networks can reduce this complexity. The author needs to dedicate space in the main text to properly explain how the discussion can be extended to $S_n \times O(d)$, perhaps drawing inspiration from [3] or [4].

6. Logical issue. In Section 3, lines 226-227, the fact that $T_L$ is $G$-equivariant and $\Phi^{(L)}$ is $O(d)$-invariant does not guarantee that the result is $G$-invariant after symmetric message aggregation. This step requires $\Phi^{(L)}$ to be $S_n$-equivariant.

The above weaknesses severely hinder the reader's ability to understand the paper. In my opinion, the author is not prepared for the paper to be accepted, and it cannot be accepted by the conference.

Reference:

[1] Satorras, Vı́ctor Garcia, et al. "E(n) Equivariant Graph Neural Networks."

[2] Thomas, Nathaniel, et al. "Tensor Field Networks: Rotation- and Translation-Equivariant Neural Networks for 3D Point Clouds."

[3] Villar, Soledad, et al. "Scalars are universal: Equivariant machine learning, structured like classical physics."

[4] Dym, Nadav, and Haggai Maron. "On the Universality of Rotation Equivariant Point Cloud Networks"

**Questions:**

See weaknesses.

---

### Official Review · Reviewer_h4h2 · 2025-10-30

**Soundness:** 3
**Presentation:** 2
**Contribution:** 2
**Rating:** 2
**Confidence:** 3

**Summary:**

This paper presents a gauge-theoretic view of transformers. By considering the map from parameters to functions, the authors derive geometric statements for transformers on the generic stratum. These statements include a characterization of the (maximal) symmetries, decomposition of the tangent space into vertical and horizontal components with respect to the gauge orbits, and statements about the differences, in terms of function, of MLP and attention layers.

**Strengths:**

The analyses are relatively straightforward and easy to follow. The symmetry group is natural and consistent with intuition and practice and overall there is a logical structure to the paper. The authors also consider practical implementations of transformers (such as RoPE and MHA), which is a nice touch.

**Weaknesses:**

As someone familiar with LLMs, group theory, and gauge theory, I found the paper difficult to read. Many things are undefined and name dropped, which hurts exposition, especially for a general audience.

While the analysis is interesting, as the authors themselves acknowledge, it is limited and restricted to a general stratum where all matrices maximally span their dimensions. Within that context, it's not clear what the practical benefit of the diagnostics is: if it is to detect whether or not we are in a general stratum regime, there are simpler tests to verify that.

Finally, the experimental section is very limited. The first experiment is not related to the algorithms presented and simply verifies the symmetry group, and algorithm 2 is not verified, which I was looking forward to.

**Questions:**

1. I struggle to understand exactly why a gauge-theoretic framework is necessary in this exposition and why standard ideas from group theory don't suffice. In that same vein, are the gauges that are considered essentially elements of $G_{\textrm{max}}$? If not, what are the gauge transformations?
2. I found $G_3$ very difficult to understand. Overall I think the paper could greatly benefit by some intuitive figures.
3. In the proofs of Theorem 2.2, why a one-parameter subgroup is considered? What are $X_i$ and $Y_i$ referenced in Lema A.3?
4. The notation $G_{\theta\mid \mathcal{H}_{\theta}}$ is introduced as the Fischer information, but that is never defined. Same thing for the Riesz representative, and the Ehresmann connection.
5. $g_{\theta}$ is only defined in reference to $g_{\theta}$-orthogonality, and the meaning isn't clear. From the context it is unclear if it is a group element (which, if it is, could be problematic since it is also treated as a metric).
6. In the same vein, line 256 talks about $g_{\theta}$-orthonormal vectors. What does that mean? Since these vectors live in $\mathcal{H}_g$ (which is orthogonal to the vertical space), isn't that a trivial statement?
7. The computation of a vertical basis is unclear. Would one have to compute the kernel of $\pi$ at $\theta$ and then span that space?
8. Morse-Bott (and all related discussions) are never defined or explained.

---

### Official Review · Reviewer_kkvL · 2025-10-31

**Soundness:** 2
**Presentation:** 1
**Contribution:** 2
**Rating:** 2
**Confidence:** 2

**Summary:**

This paper introduces a novel and theoretically deep framework for graph neural networks based on gauge theory. The central idea is to equip a graph with node features $h_i \in \mathbb{R}^d$ and "link variables" $U_{ij} \in O(d)$ on directed edges, which act as parallel transporters. The model is designed to be invariant to both node permutations ($S_n$) and local gauge transformations ($O(d)$ action on nodes, conjugation on links). The paper's main contribution is the construction of a finite, explicit dictionary of gauge invariants (open strings and Wilson loops). The authors prove, using the First Fundamental Theorem for $O(d)$, that this dictionary generates all $O(d)$-invariant polynomials, which in turn provides a universal approximation guarantee for continuous invariant functions on compact sets. The model is evaluated on a synthetic "gauge diagnostic" task and on a small subset of the QM9 dataset.

**Strengths:**

1. This paper provides a strong, principled, and novel theoretical foundation for invariant GNNs by drawing a direct connection to gauge theory and classical invariant theory.

2. The proof that a finite dictionary of "open strings" and "Wilson loops" is sufficient for universal approximation of $O(d)$-invariant functions is a significant theoretical contribution.

3. The model is provably $S_n \times O(d)$ invariant by construction, which is a much stronger and more desirable property than simple empirical invariance.

**Weaknesses:**

1. The paper fails to demonstrate a single compelling, non-synthetic use case for its framework. The only successful validation (Sec 5.2) is on a synthetic task designed by the authors to test the model's exact properties. This shows the theory works, but provides no evidence of its practical utility.

2. The attempt to use a standard benchmark (QM9) is critically flawed and unconvincing. The authors use a tiny, non-standard 3k-sample split, which makes comparison to the vast literature on QM9 impossible. Furthermore, they provide no baseline comparisons (e.g., against DimeNet, EGNN), instead only comparing their model to an augmented version of itself.

3. The paper's QM9 methodology is critically under-explained, with the core logic for data processing omitted from the paper and relegated to the supplementary code. This analysis reveals a complex, non-trivial feature engineering pipeline: the $U_{ij}$ links are derived from local PCA frames of k-nearest neighbors, and the node features $h_i$ are set to the 3D coordinates $x_i$ themselves. This convoluted setup (feeding coordinates in two redundant forms) makes the lack of baselines even more problematic.

4. The "Gauge-GNN+" model, which achieves the (un-benchmarked) "better" results on QM9, is augmented with standard atomic numbers ($Z_i$) and distances ($d_{ij}$). This suggests that the performance, such as it is, likely comes from these standard features, further obscuring any potential contribution from the novel gauge-theoretic framework.

**Questions:**

1. Why was a non-standard 3k-sample split of QM9 used instead of benchmarking against standard models (like DimeNet or EGNN) on an established split? This decision makes the model's performance on this task uninterpretable.

2. How much of the performance improvement in "Gauge-GNN+" is attributable to the standard distance/atom-type features versus the gauge-invariant features? An ablation study removing the $Z_i$ and $d_{ij}$ inputs is needed.

3. Given that the model is designed for gauge invariance, can the authors provide even one real-world, non-synthetic problem or dataset (other than QM9) where this framework is a natural fit and provides a clear advantage?

---

### Official Review · Reviewer_7Lbr · 2025-10-31

**Soundness:** 2
**Presentation:** 3
**Contribution:** 2
**Rating:** 2
**Confidence:** 4

**Summary:**

The paper introduces a gauge‑theoretic framework for GNNs where each directed edge carries a link variable $\(U_{ij}\ in O(d)\)$ that parallel‑transports node features, and the invariant head aggregates a finite dictionary of gauge invariants formed by $\{open strings} \(\langle h_i, W h_j\rangle\)$ and $\{Wilson loops} \(\operatorname{tr}(W)\)$.
Using the First Fundamental Theorem (FFT) for $\(O(d)\)$ on mixed tensors and matrix trace identities, the authors prove this dictionary generates all \(O(d)\)-invariant polynomials on the joint space of node features and link variables, yielding a universal approximation property (UAP) on compact sets; after index symmetrization, this extends to $\(S_n\times O(d)\)$.
They formalize learning on the orbit space $\(X/(S_n\times O(d))\)$ and prove nonuniform learnability via bounded‑Lipschitz slices.

**Strengths:**

1. An explicit, finite dictionary ($open strings \(\langle h_i,Wh_j\rangle\$) and Wilson loops $\(\operatorname{tr}(W)\))$ that generates $\(P(X)^{O(d)}\)$ by FFT and trace identities; bounded word lengths via Cayley–Hamilton/Razmyslov–Procesi.
2. Practical architecture: A lightweight message‑passing pipeline with a compact invariant readout $\([S_0,S_1,W_3,E_3]\)$ feeding an MLP; suitable for modest compute.

**Weaknesses:**

1.  Experiments are $\(O(d)\)$-only (mainly \(d=3\)) and small‑scale; extension to $\(SO(3)\)$, larger datasets, or other continuous groups would strengthen generality claims.
2. Expressivity vs. efficiency: The finite dictionary is attractive, but computing loops and long words can be costly on large graphs; the paper mentions sampling strategies as future work but lacks empirical cost analysis.
3. Baselines: QM9 results lack direct comparisons to strong equivariant baselines under the same split/protocol, making performance positioning difficult.
4. The holonomy energy $\(E_3\)$ is an engineered readout feature (outside the minimal invariant dictionary); its impact on results needs clearer quantification.

**Questions:**

1. Theorem 4.1 states bounded word lengths depending on $\(d\)$. Please quantify the bound and clarify the practical implications for the number of loop terms on typical graphs.
2. Orbit‑space metric: You refer to a compatible quotient metric or the pseudo‑metric induced by the invariant dictionary for Lipschitzness (Lemma C.9). Please explicitly define the metric used in practice and discuss sensitivity to this choice.
3. Role of $\(E_3\)$: Since $\(E_3\)$ is an engineered (non‑dictionary) feature, provide ablation results with/without $\(E_3\)$ on QM9 to quantify its optimization benefit.
4. Loop selection \& scaling: For larger graphs, how are loop lengths/subgraph samples selected? Provide runtime/memory numbers and an ablation over loop lengths or sampling budgets.
5. Comparisons to SOTA: Add competitive equivariant baselines under the same compact split/protocol to contextualize QM9 gains.

---

### Meta-Review · Area_Chair_ybG3 · 2025-12-30

**Summary:**

The reviewers unanimously suggested rejecting the paper (four reviewers gave it a 2 and one reviewer a 4).
The authors did not respond to the reviewers.

**Reviewer Concerns:**

NA (no rebuttal is provided).

**Reviewer Scores:**

The reviewer scores wouldn't have changed.

---

### Decision · Program_Chairs · 2026-01-26

Reject